# Dynamic Strain Measurement of Rotor Blades in Helicopter Flight Using Fiber Bragg Grating Sensor

**DOI:** 10.3390/s23156692

**Published:** 2023-07-26

**Authors:** Honglin Zhang, Zefeng Wang, Fei Teng, Pinqi Xia

**Affiliations:** 1Institute of Flight Test Technology, Chinese Flight Test Establishment, Xi’an 710089, China; honglin_cfte@163.com (H.Z.); tengfei19830567@163.com (F.T.); 2College of Aerospace Engineering, Nanjing University of Aeronautics and Astronautics, Nanjing 210016, China; xiapq@nuaa.edu.cn

**Keywords:** helicopter, rotor blade, dynamic strain measurement, fiber Bragg grating, flight

## Abstract

Fiber Bragg grating (FBG) sensor has a lot of advantages over the resistance strain gauge and has been used in many applications. However, there are few applications of rotor blade dynamic measurement in helicopter flight. In this paper, a method for blade dynamic strain measurement using an FBG sensor in a helicopter’s real flight is presented. The corresponding measurement system is established and can eliminate the effects of the helicopter’s electromagnetic environment on the electrical sensing components in the measurement system with the orthogonal frequency-division multiplexing modulation. The measured dynamic strains on the rotor blades of the helicopter in real flight contain six harmonic frequencies with the vibration characteristics of rotor blades, indicating that the established FBG measurement method and system have practical engineering applications.

## 1. Introduction

Rotor blade strain measurement is a necessary project for each helicopter in flight tests. The structural strain measurement can realize safety monitoring during the flight envelope expansion and obtain the structural load of the blade through the strain–load equation, which can check and evaluate the strength of the blade. At present, the rotor blade strain measurement is realized using resistance strain gauge measurement in which a large number of strain gauges need to be pasted on the blade surface with a large number of test wires which typically change the aerodynamic shape of the rotor blades and affect the aerodynamic efficiency and aeroelastic response characteristics of the rotor during the flight test. In addition, a resistance strain gauge has a low fatigue life which cannot meet the cycles of strain measurement in the flight test and is particularly vulnerable to electromagnetic interference which may affect the accuracy of the measurement results.

A fiber Bragg grating (FBG) sensor, as an optical signal sensor, is essentially different from a resistance strain gauge, and it has a lot of advantages such as a long-term stability, a strong corrosion resistance, a small volume, a light weight, a high sensitivity, a strong resistance to electromagnetic interference, a broad transmission band, a large communication capacity, a low transmission loss, insulation, no electric spark, small leaks, and a strong confidentiality; it has been widely used in many fields, such as: a bridge motion system for measuring the vehicle parameters based on strain measurement using FBG sensors [1], an FBG sensing application in an arch dam model test [2], and the monitoring of a coal mine roadway roof based on FBG displacement sensors [3] in the field of civil engineering; fault detection of an FBG sensing system on airplanes [4], landing-gear load measurement using a wide-range FBG strain sensor [5], border detection application via temperature, strain, and pressure using FBG [6], and FBG sensors for aircraft wing-shape measurement [7] in the field of airplane engineering; FBG data treatment aimed at eliminating the noise in the strain sensor data induced by vibrations of the helicopter blade in flight conditions [8], helicopter bearing detection technology using FBG acoustic emission technology [9], and blade vibration load characteristic measurement using FBG by the authors [10] in the field of helicopter engineering; and FBG-based acoustic emission measurement [11] and a laser self-mixing FBG sensor for acoustic emission measurement [12] for structural health monitoring.

According to the literature, there is currently very little use for FBG technology in the blade dynamic measurement of a helicopter in real flight. In this paper, a method for blade dynamic strain measurement with an FBG sensor in helicopter flight is presented and an FBG dynamic strain measurement system for the rotor blade of a helicopter in real flight is established. The strain data transmission on the rotor hub to the cabin is achieved by upper and lower communication modulation units with power line communication (PLC) technology [13]. Orthogonal frequency-division multiplexing (OFDM) [14] modulation technology is adopted to eliminate the effects of the helicopter’s electromagnetic environment on electrical sensing components. The FBG measurement system of the blade dynamic strain is tested on a helicopter rotor’s ground turntable before being used in helicopter flight. The measurement results indicate that the established FBG measurement method and system can be used to measure the dynamic strains of the helicopter rotor blade in flight.

## 2. Blade Dynamic Strain Measured by Fiber Bragg Grating

FBG sensor technology is based on the Bragg grating effect to sense the external temperature, strain, and other physical quantities. When an external light source is incident on the Bragg grating, the light with a specific wavelength is reflected due to the change in the refractive index. The rest of the unreflected light continues to travel. The reflection’s center wavelength of the light on the Bragg grating $\lambda_{B}$ is
(1)$$\lambda_{B}{=2n}_{\mathrm{eff}}\Lambda$$
where $n_{\mathrm{eff}}$ denotes the effective refractive index of the fiber core region and Λ denotes the period of the Bragg grating.

When external physical quantities such as temperature, strain, and so on cause a change in the refractive index or period of the Bragg grating, the center wavelength drifts, and the offset of the center wavelength is
(2)$${\Delta \lambda }_{B}{=2\Delta n}_{\mathrm{eff}}{\Lambda +2n}_{\mathrm{eff}}\Delta \Lambda$$

When the Bragg grating is subjected to an external strain, whether compressed or stretched, the effective refractive index and the period of the Bragg grating is changed and also varies with the temperature. The relationships between the center wavelength offset ${\Delta \lambda }_{B}$ of the Bragg grating and the strain ε and temperature ΔT are expressed as
(3)$${\Delta \lambda }_{B}{=\lambda }_{B}\left[\left({1-p}_{e}\right)\varepsilon +\left(\alpha +\xi \right)\Delta T\right]$$
(4)$$p_{e}=\left(\frac{n_{\mathrm{eff}}^{2}}{2}\right)\left[p_{12}-\nu \left(p_{11}{+p}_{12}\right)\right]$$
where α represents the thermal expansion coefficient of the fiber, ξ represents the thermal optical coefficient of the fiber material, $p_{e}$ represents the elastic optical coefficient of the fiber, ν represents the Poisson ratio, $p_{11}$ and $p_{12}$ represent the Punk coefficients. Based on Equations (3) and (4), the strain ε can be expressed as
(5)$$\varepsilon =\frac{1}{{1-p}_{e}}\frac{{\Delta \lambda }_{B}}{\lambda_{B}}{-\varepsilon }_{T}$$
(6)$$\varepsilon_{T}=\frac{\left(\alpha +\xi \right)}{{1-p}_{e}}\Delta T$$
where $\varepsilon_{T}$ represents the strain caused by the temperature. Therefore, in the measurement of the structural strain, it is necessary to install a temperature sensor at the appropriate position of the structure to compensate for the temperature strain.

For the commonly used germanium-doped quartz fiber, the relevant parameters are $p_{11}=0.121$, $p_{12}=0.27$, ν=0.17, $n_{\mathrm{eff}}=1.456$, $\xi =7.0\times 10^{-6}/{}^{\circ}C$, and $\alpha =5.0\times 10^{-7}/{}^{\circ}C$. Hence, the parameter $p_{e}=0.22$ can be obtained by expression (4).

The vibration of the rotor blade has a typical periodic characteristic under the periodic aerodynamic force in helicopter flight and causes a periodic dynamic strain on the blade surface. With FBG sensor technology, the strain is transmitted to the FBG sensor in the form of a strain wave through the elastic material and adhesive layer of the test structure. From the transmission process of the strain wave, the main factors contributing to the frequency response characteristics of the FBG sensors are the length of the FBG sensors and the travel speed of the strain wave in the test structure material.

Due to the periodicity of the blade’s flapping motion, the strain wave on the blade surface can be considered as a periodic sinusoidal wave with a harmonic frequency. The response characteristic of the FBG sensor to the sine wave ${\varepsilon =\varepsilon }_{0}\sin(2\pi x)/\lambda$ is shown in Figure 1 where $\varepsilon_{0}$ represents the peak of the strain wave, λ represents the wavelength of the strain wave, and $l_{0}$ represents the package length of the FBG sensor. Then, the coordinates on either side of the package length are $x_{1}=\frac{\lambda }{4}-\frac{l_{0}}{2}$ and $x_{2}=\frac{\lambda }{4}+\frac{l_{0}}{2}$.

The averaged strain $\varepsilon_{p}$ measured by the FBG sensor is expressed as
(7)$$\varepsilon_{p}=\frac{\int_{x_{1}}^{x_{2}}\varepsilon_{0}\sin\frac{2\pi }{\lambda }\mathrm{xdx}}{x_{2}{-x}_{1}}=\frac{{\lambda \varepsilon }_{0}}{{\pi l}_{0}}\sin\frac{{\pi l}_{0}}{\lambda }$$

The measurement error is
(8)$$e=\left|\frac{\lambda }{{\pi l}_{0}}\sin\frac{{\pi l}_{0}}{\lambda }-1\right|$$

According to Equation (8), the measurement error is related to the ratio between the strain wave’s wavelength and the FBG sensor package length, i.e., ${n=\lambda /l}_{0}$; the error decreases with the increase of n. Generally, n is taken within 10~20.

For the strain wave’s wavelength of the test structure material, there is the following relationship:(9)$$\lambda =\frac{v}{f}$$
where f is the measurement frequency of the FBG sensor, and v is the travel velocity of the strain wave in the measured structure material. There is also the following formula for FBG sensors:(10)$${\lambda =\mathrm{nl}}_{0}$$

Substituting Equation (10) into Equation (9), we obtain
(11)$$f=\frac{v}{{\mathrm{nl}}_{0}}$$

For modern helicopters, the main and tail rotor blades are fully composite material structures, and the skin material of the blade is glass cloth. Table 1 lists the highest working frequency of an FBG sensor on glass for v=1500 m/s. It can be seen that when the package length ranges from 20 mm to 40 mm, and n is 10 and 20, respectively, the highest identification frequency range of the FBG sensor is 1880~7500 Hz. That is to say, the FBG sensor has a good applicability as long as the vibration frequency of the measured structure is less than that frequency range. Generally, a modern helicopter’s main rotor blade vibration frequency is less than 30 Hz, and the tail rotor blade vibration frequency is less than 100 Hz, which meet the above frequency range requirements. Therefore, an FBG sensor has excellent applicability in rotor blade dynamic strain measurement.

The wavelength allocation of FBG according to the blade dynamic strain measurement ranges is listed in Table 2. There are four locations of dynamic strains on the blade that need to be measured, the dynamic strain ranges are all ±3500με, and the sampling rates are all 1024 Hz. According to the blade dynamic strain measurement ranges and sampling rates listed in Table 2, each FBG wavelength ranges are listed in Table 3 where the sensor at 0.150R is for the temperature compensation. The FBG demodulator used for the helicopter flight measurement was the H19, made by Micron Optics, and its wavelength range was from1510 nm to 1560 nm.

## 3. FBG Measurement System of Blade Dynamic Strain

The design scheme of the blade dynamic strain FBG measurement system is shown in Figure 2 and consists of three parts: the optic fiber and the FBG pasted on the rotor blade surface, the upper acquisition module installed on the rotor hub, and the data recording module installed in the helicopter cabin. The upper acquisition module includes the upper communication modulation unit, azimuth measurement unit, GPS clock unit, and FBG demodulator. The GPS clock unit is used to provide the GPS clock signal. The azimuth measurement unit is used to measure the blade azimuth signal during the rotor’s rotation. The FBG demodulator is used to output and receive the optical signal from the FBG sensors and transform the optical signal into an electrical signal. The measured strain signal, GPS clock signal, and azimuth signal are modulated and transmitted by the upper communication modulation unit and delivered to the data recording module installed in the helicopter cabin through slip-ring wire 2 by using PLC technology. The upper acquisition module is powered through slip-ring wires 1 and 2.

The whole blade dynamic strain measurement system includes not only the fiber optic sensing and measurement components but also many electrical components such as the sensor end and signal transmission link. These electrical components are affected by electromagnetic interference, which affects the measurement accuracy. Therefore, it is necessary to eliminate electromagnetic interference by using PLC technology.

Common PLC technology uses a single-frequency carrier modulation for signal transmission, and the disadvantage of this modulation method is its poor anti-interference ability, which can easily cause a signal transmission failure. To solve this problem, the orthogonal frequency-division multiplexing (OFDM) modulation method was adopted. The basic OFDM principle is based on the multicarrier transmission technology. The carrier with a bandwidth B is divided into N orthogonal subcarriers with a bandwidth Δf=B/N. If the center frequency point of the first carrier is $f_{0}$, then the frequency point of the $n^{\mathrm{th}}$ carrier is $f_{0}+\left(n-1\right)\Delta f$. Modulating the symbol $X_{n}$ to the $n^{\mathrm{th}}$ carrier to obtain the transmitted symbol $X_{n}e^{{j2\pi f}_{n}t}$, the final transmitted signal $f\left(t\right)=\sum_{n=1}^{N}X_{n}e^{{j2\pi f}_{n}t}{=e}^{{j2\pi f}_{0}t}\sum_{n=1}^{N}X_{n}e^{j2\pi n\Delta \mathrm{ft}}$ can be found after the signals on all N carriers are accumulated.

Following the reception of the signal $f\left(t\right)$, the symbol $X_{n}$ transmitted on the $n^{\mathrm{th}}$ carrier can be determined by
(12)$$\Delta f\int_{0}^{1/\Delta f}f\left(t\right)e^{{-j2\pi f}_{n}t}{\mathrm{dt}=X}_{n}+\sum_{k¹n}\Delta f\int_{0}^{1/\Delta f}X_{k}e^{{j2\pi f}_{k}t}e^{{-j2\pi f}_{n}t}$$
where $f\left(t\right){=e}^{{j2\pi f}_{0}t}\sum_{n=1}^{N}X_{n}e^{j2\pi n\Delta \mathrm{ft}}$ is the further carrier modulation of the baseband signal $\hat{f}\left(t\right)=\sum_{n=1}^{N}X_{n}e^{j2\pi \Delta \mathrm{ft}}$ before transmission. When $\hat{f}\left(t\right)$ is a continuous signal, the time interval T=1/B is adopted to sample the signal, and the first sample is $\hat{f}\left(1T\right)=\sum_{n=1}^{N}X_{n}e^{j2\pi \frac{\mathrm{nl}}{N}}$, i.e., the first point in the sequence $\left\{\begin{array}{cccc} \hat{f}\left(1\right) & \hat{f}\left(2\right) & \cdots & \hat{f}\left(N\right) \end{array}\right\}$ obtained after the inverse discrete Fourier transform (IDFT) is applied to the signal sequence $\left\{\begin{array}{cccc} X_{1} & X_{2} & \cdots & X_{n} \end{array}\right\}$. In OFDM, N sampling points of the continuous signal $\hat{f}\left(t\right)$ are actually transmitted. When the receiver $\left\{\begin{array}{cccc} \hat{f}\left(1\right) & \hat{f}\left(2\right) & \cdots & \hat{f}\left(N\right) \end{array}\right\}$ applies a discrete Fourier transform (DFT) to these N points, the column data $\left\{\begin{array}{cccc} X_{1} & X_{2} & \cdots & X_{n} \end{array}\right\}$ can be finally obtained. The schematic diagram of signal transmission based on OFDM modulation is shown in Figure 3. 

## 4. Validation of Measurement Method and System

Before the real flight measurement, the blade dynamic strain FBG measurement system was tested by a ground turntable measurement. Figure 4 shows a photo of the blade dynamic strain FBG measurement system installed on a helicopter rotor’s ground turntable. As shown in Figure 4, the optical fiber indicated by an arrow was pasted on the blade surface, and the upper acquisition module, including the upper communication modulation unit, azimuth measurement unit, GPS clock unit and the FBG demodulator as shown in Figure 2 were installed inside the white cylindrical cover indicated by an arrow. The optical fiber connected with the FBG demodulator. The dynamic strains at four locations on the blade listed in Table 2 under different pitch angles of the blade were measured. The original FBG spectrum and the induced dynamic strain FBG spectrum at the second location of the blade when the rotation speed was 100 rpm are shown in Figure 5. By comparison, it can be seen that the shapes of the two curves are identical, but the dimensions of the two curves are different. Hence, the original FBG spectrum reflects the dynamic strain characteristics of the blade, and the measurement system can be used to measure the dynamic strains of the blade. The measured dynamic strain of the rotor blade in the frequency domain is shown in Figure 6, which includes the first three harmonic vibration modes of the blade.

In order to validate the measurement accuracy of the blade dynamic strain FBG measurement system, the strain measurement of an equal-strength beam under a force was taken as shown in Figure 7 where H, L, and B are the thickness, length, and end width of the beam, respectively. A force F was applied at the free end of the beam. The FBG sensors and strain gauges were pasted on two sections of the beam to measure the strains under the force F. The measured strains and the errors by the FBG sensors and strain gauges are listed in Table 4. It can be seen that all the errors were very small, and the maximum error was only 1.8%, indicating that the measurement accuracy of the blade dynamic strain FBG measurement system was very high for an engineering application.

## 5. Measured Dynamic Strains of Rotor Blade in Real Flight

In the flight test, the steady-state flight was selected to measure the strains of the rotor blades. Figure 8 shows the time-domain curves of the measured dynamic strains at each blade location when the helicopter was in hover. Figure 9 shows the frequency spectrum of the measured dynamic strains at each location of the blade in hover and the first six harmonic frequencies at each blade location. It should be noted that when r = 0.196R, the amplitude of the fifth harmonic frequency was much larger than that of the first harmonic frequency because the fifth harmonic frequency 5Ω (Ω is the rotor rotational speed) of the rotor centrifugal force was close to the fifth harmonic frequency of the blade and 0.196R was located at the higher position of the fifth harmonic mode shape of the blade. Figure 10 shows the frequency-domain results of the measured dynamic strains when the helicopter was in a steady-state forward flight, which included the first six harmonic frequencies at each blade location. Figure 11, Figure 12 and Figure 13 show the measured dynamic strains’ amplitudes at each harmonic frequency varied with the advance ratios when the FBG sensors were, respectively, at blade locations 0.015R, 0.196R, and 0.445R in forward flight. It can be seen from Figure 11, Figure 12 and Figure 13 that the measured dynamic strains’ amplitudes of the first harmonic frequency spectrum at three blade locations were all large in forward flight and increased with the increasing advance ratio. The amplitudes of the second harmonic frequency spectrum at three blade locations obviously increased with the increasing advance ratio. The amplitudes of the third and fourth harmonic frequency spectrum at three blade locations increased first and then decreased with the increasing advance ratio. The amplitudes of the fifth and sixth harmonic frequency spectrum at three blade locations were smaller than the amplitudes of the first four harmonic frequency spectrums and decreased with the increasing advance ratio.

## 6. Conclusions

The blade dynamic strain FBG measurement method and system during a helicopter’s real flight established in this paper was applied in practice and had a strong resistance to electromagnetic interference and a good measurement accuracy. The measured dynamic strains on the rotor blades during a helicopter’s real flight contained six harmonic frequencies which reflected the vibration characteristics of the rotor blades. According to the measured dynamic strains on the blades in the helicopter’s forward flight, the amplitudes of the first harmonic frequency spectrum at different locations on the blade were all large and increased with an increasing advance ratio. The amplitudes of the second harmonic frequency spectrum obviously increased with an increasing advance ratio. The amplitudes of the third and fourth harmonic frequency spectrums increased first and then decreased with the increasing advance ratio. The amplitudes of the fifth and sixth harmonic frequency spectrums decreased with the increasing advance ratio.

## Figures and Tables

**Figure 1 sensors-23-06692-f001:**
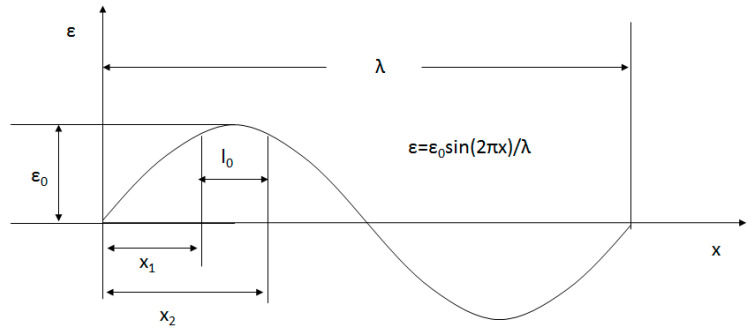
Response characteristics of an FBG sensor to a sinusoidal strain wave.

**Figure 2 sensors-23-06692-f002:**
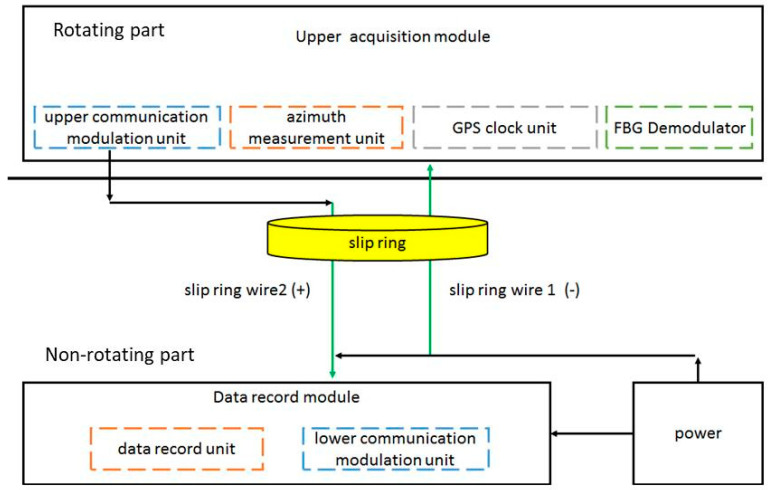
Design scheme of blade dynamic strain FBG measurement system.

**Figure 3 sensors-23-06692-f003:**
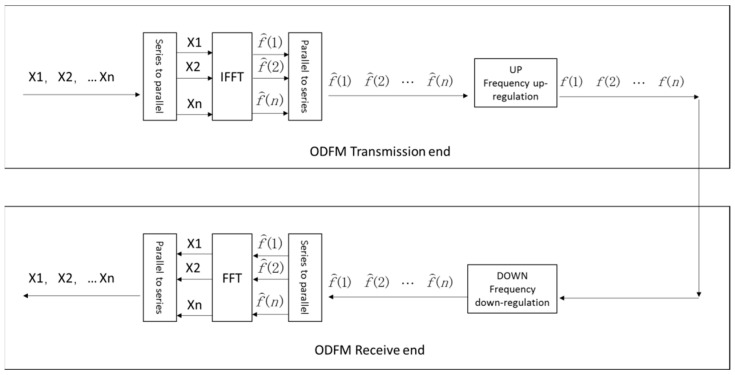
Schematic diagram of signal transmission based on OFDM modulation.

**Figure 4 sensors-23-06692-f004:**
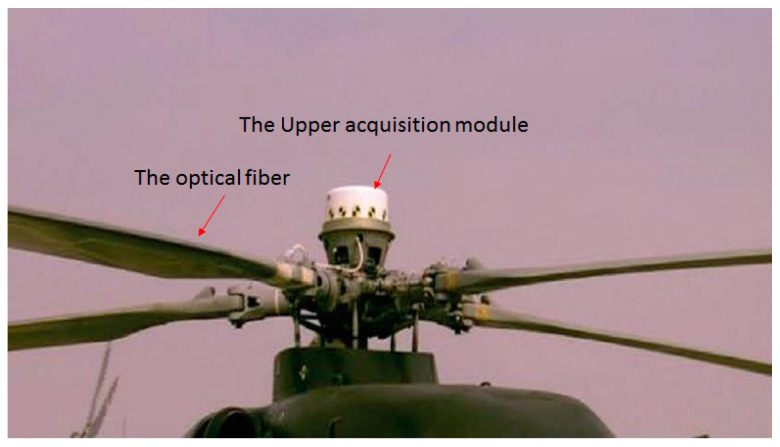
Blade dynamic strain FBG measurement system installed on a helicopter rotor’s ground turntable.

**Figure 5 sensors-23-06692-f005:**
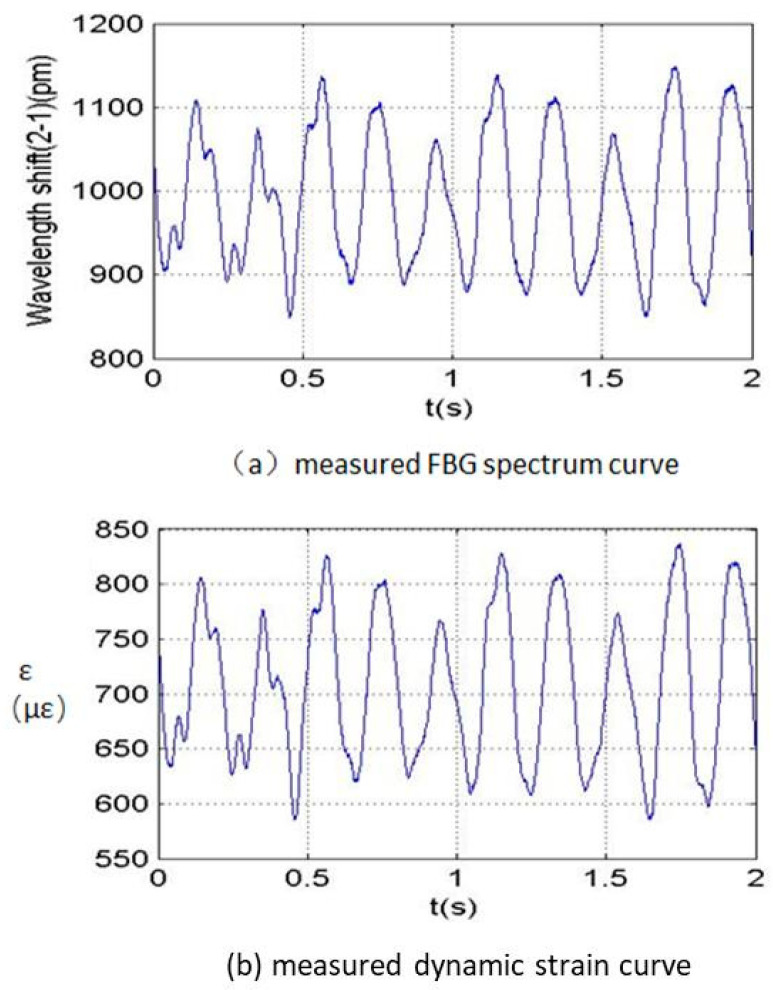
Original FBG spectrum and induced dynamic strain FBG spectrum at the 2nd location of the blade.

**Figure 6 sensors-23-06692-f006:**
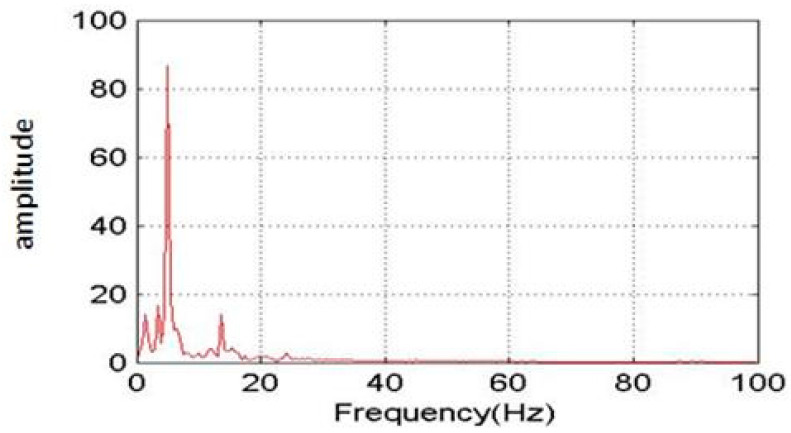
Measured dynamic strain at the 2nd location of the rotor blade in the frequency domain.

**Figure 7 sensors-23-06692-f007:**
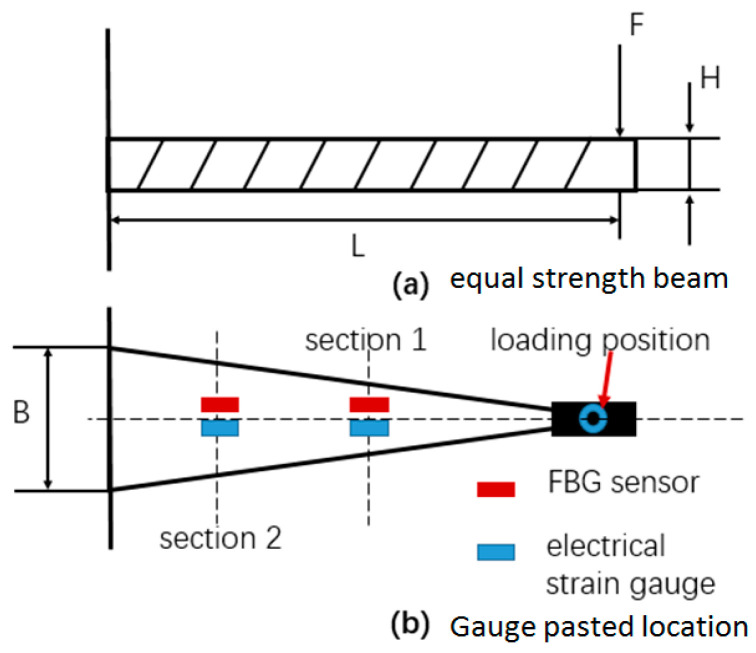
Diagram of the strain measurement of an equal-strength beam.

**Figure 8 sensors-23-06692-f008:**
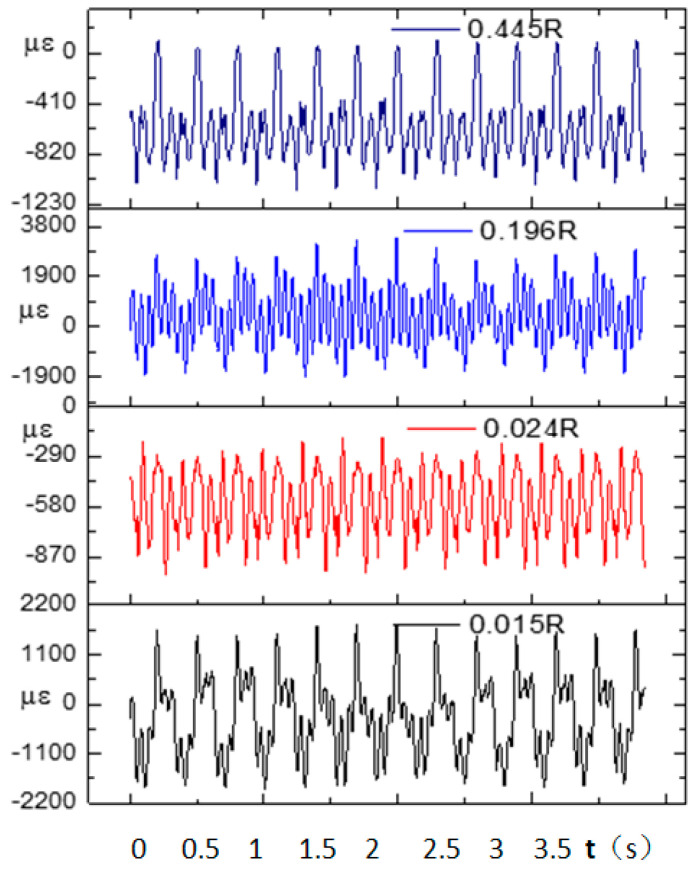
Time-domain curves of measured dynamic strains at each blade location in hover.

**Figure 9 sensors-23-06692-f009:**
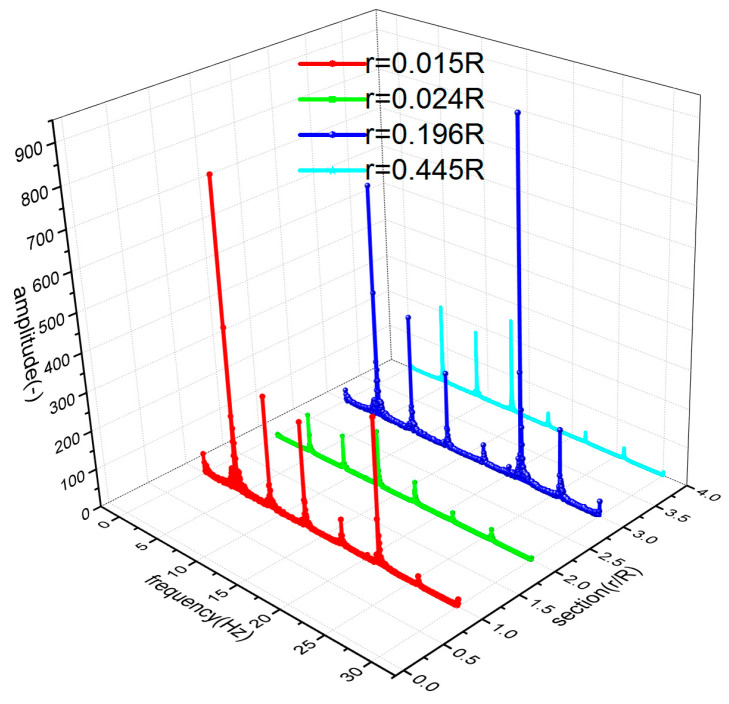
Frequency spectrum of the measured dynamic strains at each location of the blade in hover.

**Figure 10 sensors-23-06692-f010:**
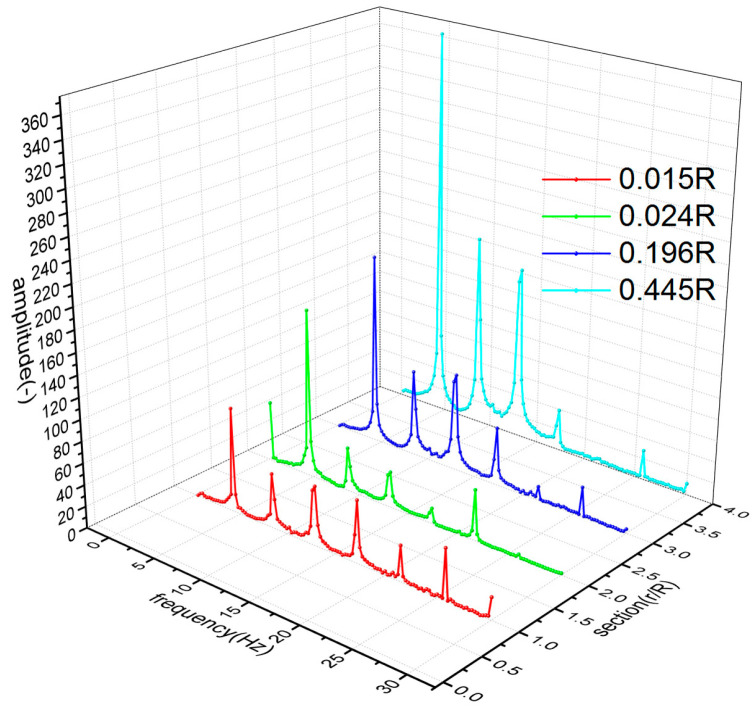
Frequency spectrum of the measured dynamic strains at each location of the blade in a steady-state forward flight.

**Figure 11 sensors-23-06692-f011:**
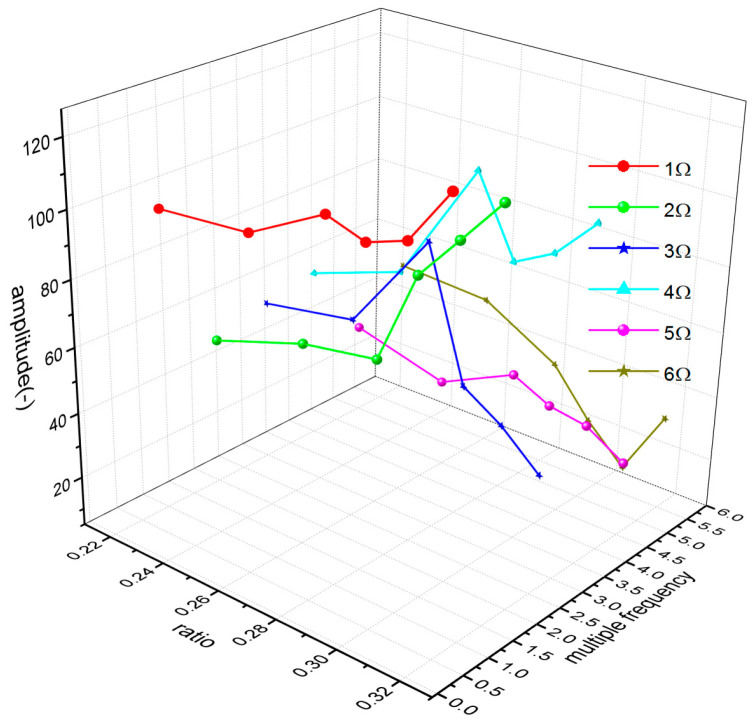
Measured dynamic strain amplitude of each harmonic frequency varies with advance ratios at the 0.015R blade location.

**Figure 12 sensors-23-06692-f012:**
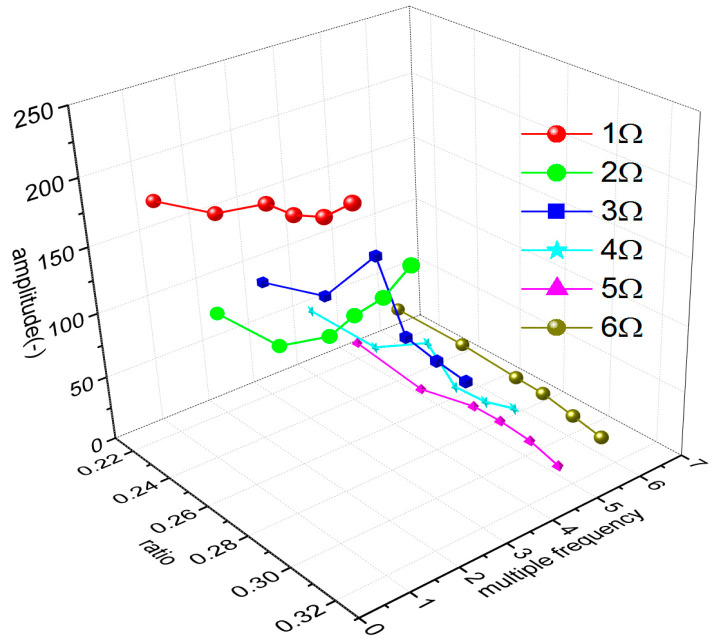
Measured dynamic strain amplitude of each harmonic frequency varies with advance ratios at the 0.196R blade location.

**Figure 13 sensors-23-06692-f013:**
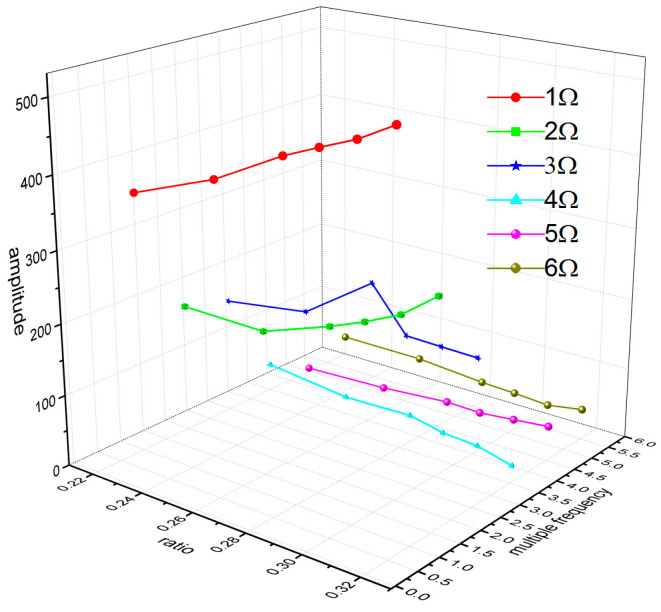
Measured dynamic strain amplitude of each harmonic frequency varies with advance ratios at the 0.445R blade location.

**Table 1 sensors-23-06692-t001:** Highest working frequency of an FBG sensor at different n and $l_{0}$.

n	l_0_ (mm)	Highest Working Frequency (Hz)
10	20	7500
	25	6000
	30	5000
	40	3750
20	20	3750
	25	3000
	30	2500
	40	1880

**Table 2 sensors-23-06692-t002:** Measurement ranges and sampling rates at four locations.

No.	Location	Dynamic Strain Range (με)	Sampling Rate (Hz)
1	0.015R	±3500	1024
2	0.024R	±3500	1024
3	0.196R	±3300	1024
4	0.445R	±3200	1024

**Table 3 sensors-23-06692-t003:** Wavelength assignments of each measurement location.

No.	Locations	Start Wavelength	End Wavelength	Central Wavelength
1	0.015R	1510 nm	1520 nm	1515 nm
2	0.024R	1520 nm	1530 nm	1525 nm
3	0.150R	1530 nm	1536 nm	1533 nm
4	0.196R	1536 nm	1546 nm	1541 nm
5	0.445R	1546 nm	1556 nm	1551 nm

**Table 4 sensors-23-06692-t004:** Measured strains by the FBG sensors and strain gauges.

F(N)	Section 1 on Equal Strength Beam			Section 2 on Equal Strength Beam		
	FBG Sensor (με)	Strain Gauge (με)	Error	FBG Sensor (με)	Strain Gauge (με)	Error
9.8	82.6	83.7	1.3%	84.6	86.6	1.1%
19.6	167.6	168.7	0.6%	171.3	174.3	1.8%
29.4	252.7	254.7	0.7%	259	261.8	1.0%
39.2	342.4	344.1	0.5%	346.4	349.3	0.8%
49.0	429.9	430.2	0.07%	433.2	437.0	0.9%
58.8	513.3	514.2	0.2%	516.7	522.9	1.2%

## Data Availability

Data sharing not applicable.

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
