# Peer review of "Dynamic Strain Measurement of Rotor Blades in Helicopter Flight Using Fiber Bragg Grating Sensor"

_sensors, 2023, doi:10.3390/s23156692_

Round 1
Reviewer 1 Report
This paper has described a method for measuring dynamic strain on helicopter blades in flight using FBG sensor. The authors have also used orthogonal frequency division multiplexing modulation to eliminate the effects of the helicopter's electromagnetic environment. The methods have potential application in practical engineering applications. There are several concerns with regard to the ability of the proposed scheme:
1. What is represented by the horizontal coordinate in Figure 7, in other words, what is its specific parameter unit?
2. In general, temperature has a significant influence on the FBG measurement process. What method did the authors use to remove the temperature crosstalk during the measurement?
3. When r = 0.196R in Figure 8, why is amplitudes of the 3rd harmonic frequency much larger than that of 1rd harmonic frequency? Because from the Fourier series spectrum analysis, the general trend is that the higher harmonics are gradually decaying, corresponding to a decreasing amplitude.
4. From the experimental results, the difference between the monitored strains at different locations is relatively large. So for this kind of structure with uneven stress distribution, whether distributed sensors are more suitable for it?
Author Response
Please see the attachment。

Reviewer 2 Report
The authors proposed to use FBGs to measure dynamic strain information in a helicopter. It is a good attempt for extending applications of FBG. Before publishing, some concerns should be addressed.
1. In the introduction, the authors said” FBG has strong resistance to electromagnetic interference”, but in line 151, they also said “to eliminate the effects of electromagnetic environment of the helicopter”. Is it self-contradictory? Can authors explain it?
2. In line 93, the authors said “Due to the periodicity of the blade flapping motion, the strain wave on the blade 93 surface can be considered as a periodic sine wave”. Why it can be considered as a periodic sinusoidal wave? Based on the experimental results, it seems not a sinusoidal wave?
3. In Figure 4, authors can indicate the locations of FBG, and strain modulator and how they connected like the circuit routine.
4. When review FBGs in the introduction, some references can be added like:
(a) S. Jinachandran, and G. Rajan, "Fibre Bragg Grating Based Acoustic Emission Measurement System for Structural Health Monitoring Applications," Materials 14, 897, 2021.
(b) B. Liu, Y. Ruan, Y. Yu, J.i, Q., J., and G. Rajan, “Laser self-mixing fiber Bragg grating sensor for acoustic emission measurement”, Sensors, 18(6), 1956, 2018.
The English should be polished before publishing.
Reviewer 3 Report
This manuscript shows the dynamic strain measurement for rotor blades of helicopter flight by using fiber gratings. My comments are following:
1. It is a nice design by using a fiber Bragg grating sensor to real-time measure the dynamic strain in rotor blades of helicopter in real flight.
2. Authors should clearly show the dynamic strain measurement configuration shown in Fig. 4. The original FBG spectrum and the induced dynamic strain FBG spectrum are also required to be shown and described.
3. Authors should describe the thermal effect for measuring the dynamic strain in rotor blades.
4. For the conclusions, the two paragraphs combine together to a paragraph and to remove (1) and (2).
I think after minor revision this manuscript is suitable to be published in Sensors.
This manuscript shows the dynamic strain measurement for rotor blades of helicopter flight by using fiber gratings. My comments are following:
1. It is a nice design by using a fiber Bragg grating sensor to real-time measure the dynamic strain in rotor blades of helicopter in real flight.
2. Authors should clearly show the dynamic strain measurement configuration shown in Fig. 4. The original FBG spectrum and the induced dynamic strain FBG spectrum are also required to be shown and described.
3. Authors should describe the thermal effect for measuring the dynamic strain in rotor blades.
4. For the conclusions, the two paragraphs combine together to a paragraph and to remove (1) and (2).
I think after minor revision this manuscript is suitable to be published in Sensors.
Reviewer 4 Report
The study does not seem to be conclusive. The bibliography is not complete and the introduction needs to be expanded. The innovation introduced by the manuscript is not clear. This method to measure the blade dynamic blade does not seem to be properly validated. The single components used and their characteristics of the measurement system were not introduced and described. The readability of the figures is difficult. The error introduced by this measurement system should be assessed and shown with regards to those systems already present in bibliography. Considering the state of the study, the manuscript is not suitable to be published on this journal.
English needs extensive review and editing. Many parts of the document are hardly comprehensible.
Round 2
Reviewer 1 Report
The revised manuscript can be accepted for publication by this journal.
Reviewer 2 Report
The authors have addressed my concerns. It can be accepted for publishing!
Minor revisions for the language are needed.
Reviewer 4 Report
The paper was greatly improved. The revised paper can be accepted for publication now.
Minor editing of English language would still be necessary to improve the manuscript even more.